# Pre-clinical investigation of astatine-211-parthanatine for high-risk neuroblastoma

Mehran Makvandi[1], Minu Samanta[2], Paul Martorano[1], Hwan Lee[1], Sarah B. Gitto[3,4,5], Khushbu Patel[2], David Groff [2], Jennifer Pogoriler [6], Daniel Martinez[6], Aladdin Riad[1], Hannah Dabagian[1], Michael Zaleski [1], Tara Taghvaee[1], Kuiying Xu[1], Ji Youn Lee[1], Catherine Hou [1], Alvin Farrel[2], Vandana Batra[2], Sean D. Carlin[1], Daniel J. Powell Jr [3,4,5], Robert H. Mach[1], Daniel A. Pryma [1✉] & John M. Maris [2,7✉]

Astatine-211-parthanatine ([$^{211}$At]PTT) is an alpha-emitting radiopharmaceutical therapeutic that targets poly(adenosine-diphosphate-ribose) polymerase 1 (PARP1) in cancer cells. High-risk neuroblastomas exhibit among the highest PARP1 expression across solid tumors. In this study, we evaluated the efficacy of [$^{211}$At]PTT using 11 patient-derived xenograft (PDX) mouse models of high-risk neuroblastoma, and assessed hematological and marrow toxicity in a CB57/BL6 healthy mouse model. We observed broad efficacy in PDX models treated with [$^{211}$At]PTT at the maximum tolerated dose (MTD 36 MBq/kg/fraction x4) administered as a fractionated regimen. For the MTD, complete tumor response was observed in 81.8% (18 of 22) of tumors and the median event free survival was 72 days with 30% (6/20) of mice showing no measurable tumor >95 days. Reversible hematological and marrow toxicity was observed 72 hours post-treatment at the MTD, however full recovery was evident by 4 weeks post-therapy. These data support clinical development of [$^{211}$At]PTT for high-risk neuroblastoma.

[1] Division of Nuclear Medicine and Clinical Molecular Imaging, Department of Radiology, University of Pennsylvania Perelman School of Medicine, Philadelphia, PA 19104, USA. [2] Division of Oncology and Center for Childhood Cancer Research, Children's Hospital of Philadelphia, Colket Translational Research Building, 3501 Civic Center Boulevard, Philadelphia, PA 19104, USA. [3] Ovarian Cancer Research Center, Division of Gynecology Oncology, Department of Obstetrics and Gynecology, Perelman School of Medicine, University of Pennsylvania, Philadelphia, PA 19104, USA. [4] Center for Cellular Immunotherapies, University of Pennsylvania, Philadelphia, PA 19104, USA. [5] Department of Pathology and Laboratory Medicine, Perelman School of Medicine, University of Pennsylvania, Philadelphia, PA 19104, USA. [6] Department of Pathology and Laboratory Medicine, Children's Hospital of Philadelphia, Philadelphia, PA 19104, USA. [7] Department of Pediatrics, Perelman School of Medicine at the University of Pennsylvania, Philadelphia, PA 19104, USA. ✉email: daniel.pryma@uphs.upenn.edu; Maris@chop.edu

Children with high-risk neuroblastoma have less than a 50% overall survival despite therapy intensification over several decades[1,2]. The rapid proliferative capacity of high-risk neuroblastomas is accompanied by overexpression of DNA damage response enzyme, poly(ADP-ribose) polymerase 1 (PARP1)[3]. Compared with other solid tumor types, we previously showed that high-risk neuroblastomas have among the highest PARP1 gene expression[3]. Given the impressive single-agent activity of beta-emitting radiopharmaceutical therapeutic – iodine-131-metaiodobenzylguanidine ([131I]MIBG), we sought to develop a PARP1 targeted radiopharmaceutical therapeutic that uses an alpha-emitting radionuclide to overcome the biophysical limitations of beta-therapy[4,5]. Indeed, overall survival benefit from [131I]MIBG remains below 40% and has been attributed to relapse caused by inefficient killing of micrometastatic tumors[4]. Unlike beta-particles that cause single-stranded DNA breaks and require thousands of decay events for cell death, alpha-particles cause complex double-stranded DNA breaks that result in cell death with as few as 5 decay events if cell nuclei are hit[5]. The extreme cytotoxicity of alpha-particles is balanced by a short-path length of 2–3 cell diameters (70 μm) that provides a mechanism to deliver radiation potent enough to kill a single cancer cell and spare nearby normal tissue[6].

By radiolabeling a small molecule PARP inhibitor with astatine-211 ($^{211}$At), we developed the first-in-class alpha-emitting drug that targets cancer nuclei via PARP1, [211At]parthanatine ([211At] PTT) – previously referred to as [211At]MM4[7]. As a therapeutic, [211At]PTT completes a theranostic pair with companion diagnostic fluorine-18 Fluorthanatrace ([18F]FTT), which is currently under evaluation in multiple adult cancers for positron-emission tomography (PET) imaging of PARP1[8–10]. Both agents are structurally most similar to the FDA-approved PARPi – rucaparib, and display similar affinity to PARP1 in vitro[11–13]. Chemically, [18F] FTT and [211At]PTT only differ by a single functional group (fluorethoxy vs. astato) where the radionuclide is incorporated. Therefore, it is expected that both agents would exhibit similar kinetics in vivo. In ovarian cancer patients, [18F]FTT displayed rapid tumor uptake kinetics between 60–180 minutes post injection with minimal wash-out[14]. Considering [18F]FTT as a surrogate for PARPi, We envisioned the rapid tumor targeting kinetics, slow target off-rate, and deep tumor penetration of a small molecule PARPi to be perfectly matched for the half-life of $^{211}$At (7.2 h). This approach maximizes tumor radiation dose possible with $^{211}$At, and due to the sub-cellular delivery in the nucleus it deposits more than 3 times the radiation dose per decay compared with cytosolic delivery of $^{211}$At[15]. Although the question of the clinical significance of targeting alpha-emitters to the nucleus remains to be unanswered, in theory it should have beneficial effects especially for cancer-specific targets. Conjugation chemistry of radiometals, such as $^{225}$Ac, requires metal chelation and is not suitable for organic small molecule targeting vectors, such as PARPi, that must diffuse across cell membranes. Unlike radiometal alpha-emitter complexes, $^{211}$At is a radiohalogen that can be covalently incorporated into small molecule drugs that provides a unique platform for alpha-therapeutics targeted to intracellular proteins. Furthermore, sufficient tumor diffusion of alpha-emitting therapeutics has been shown to be a critical component for optimal tumor response[16], and organic small molecule PARPi exhibit large volume of distribution that suggests rapid transfer of drug from circulation to peripheral tissues. In this study we seek to deliver alpha-radiation directly to cancer cell nuclei by targeting PARP1.

In early proof of concept studies we demonstrated that [211At] PTT (previously referred to as [211At]MM4) was nearly 1 billion times more cytotoxic than the non-radioactive PARP inhibitor analog (KX1)[3]. Using a single high-risk neuroblastoma tumor xenograft mouse model (IMR-05) we showed [211At]PTT was safe and effective, especially when administered using a fractionated dosing regimen[3]. Here, we sought to determine the efficacy of [211At]PTT against a broad and diverse panel of neuroblastoma patient-derived xenograft (PDX) mouse models and assessed for early toxicity in immune-competent mice.

## Results

**Preclinical efficacy study.** To rigorously test the clinical utility of [211At]PTT we evaluated efficacy in 11 randomly selected high-risk neuroblastoma PDX mouse models. Patient demographics and genetic characteristics are listed in (Supplementary Tables 1, 2)[17]. The maximum tolerated dose (MTD) was determined to be 36 MBq/kg administered by intra-peritoneal injection twice weekly for a total of 4 dose fractions and this fractionated regimen was used in all studies (Supplementary Fig. 1)[18,19]. In this study we evaluated the efficacy of [211At]PTT at the MTD and two lower dose levels (12 and 24 MBq/kg) administered as a fractionated regimen (Fig. 1a). Nine of 11 PDX models were MYCN amplified (Fig. 1b), which is a hallmark of high-risk neuroblastoma. Tumor response was measured as the maximum percent decrease in tumor volume after treatment and event-free survival (EFS) was measured by the area over the curve method with tumor events pre-defined by tumor growth exceeding 2 cm[3,18].

We observed potent anti-tumor activity across all dose levels (tumor growth curves shown as spider plots in (Supplementary Fig. 2a, b, and Supplementary Table 3). The average decrease in tumor volume for all models treated at 36 (MTD), 24, or 12 MBq/ kg dose levels were 98.4 ± 3.3% ($n = 22$; 11 PDX models), 92.6 ± 8.1% ($n = 14$, 7 PDX models), and 27.15 ± 42.2% ($n = 8$; 4 PDX models). Only two mouse tumors showed progressing disease after treatment at the 12 MBq/kg dose level (COG-N-415x; $n = 1$, and COG-N-453x; $n = 1$) (Fig. 1c). Tumor response was significantly greater for mice treated at the MTD compared to lower dose levels (paired-model T-test; p-value = * <0.05) (Fig. 1c).

Six of 20 (30%) mice treated at the MTD had no measurable disease at end of study (>90 days) and included 4 PDX models: COG-N-426x-FELIX ($n = 2$), COG-N-519x ($n = 1$), NB-1643x ($n = 1$), and NB-EBC1x ($n = 1$) (Fig. 1d). The median EFS for groups treated at the 36, 24, and 12 MBq/kg dose levels were 74, 46, and 23 days and were significantly longer for all dose levels compared to control (EFS 11 days) (paired-model ANOVA analysis; p value = *0.013, **0.0019, ****<0.0001) (Fig. 1d). Grouped analysis for models treated at all dose levels (COG-n-415x, COG-n-440x, and COG-n-519x) showed significantly longer EFS compared to controls (T-test, p value < 0.05) (SI Fig. 3).

Tumor re-challenge studies at the 24 MBq/kg dose level were performed in two models (COG-N-426x-Felix: $n = 1$, and NB-EBC1x: $n = 1$) to test if the initial tumor response was maintained after subsequent therapy. We observed 100% ($n = 2$) reduction in tumor volume after therapy re-challenge, even when tumor size was large (~1 cm$^3$) in the NB-EBC1x model (Fig. 1e). EFS between initial and re-challenge treatment were 49 vs. 59 days (COG-N-426x-FELIX, no measurable disease) and 45 vs. 47 for NB-EBC1x (progressing disease).

Two out of 22 (9%) of mice treated at the MTD were censored from EFS analysis due to weight loss >20%, albeit >50 days from treatment and both subjects had no measurable disease at time of removal (NB-1643x at 76 days, and COG-N-471x at 65 days). One dose-limiting toxicity was observed in an NE-EBC1x tumor-bearing mouse re-challenged with 24 MBq/kg administered as a fractionated regimen (Fig. 1f), and overall 43 of 46 treated mice for all dose levels maintained a healthy weight throughout the study (≥100 ± 20% body weight) (mouse weights shown in Supplementary Fig. 2c).

No associations were found between EFS and PARP1 mRNA expression (Fig. 2a) or between genetic sub-groups (Fig. 2b). In

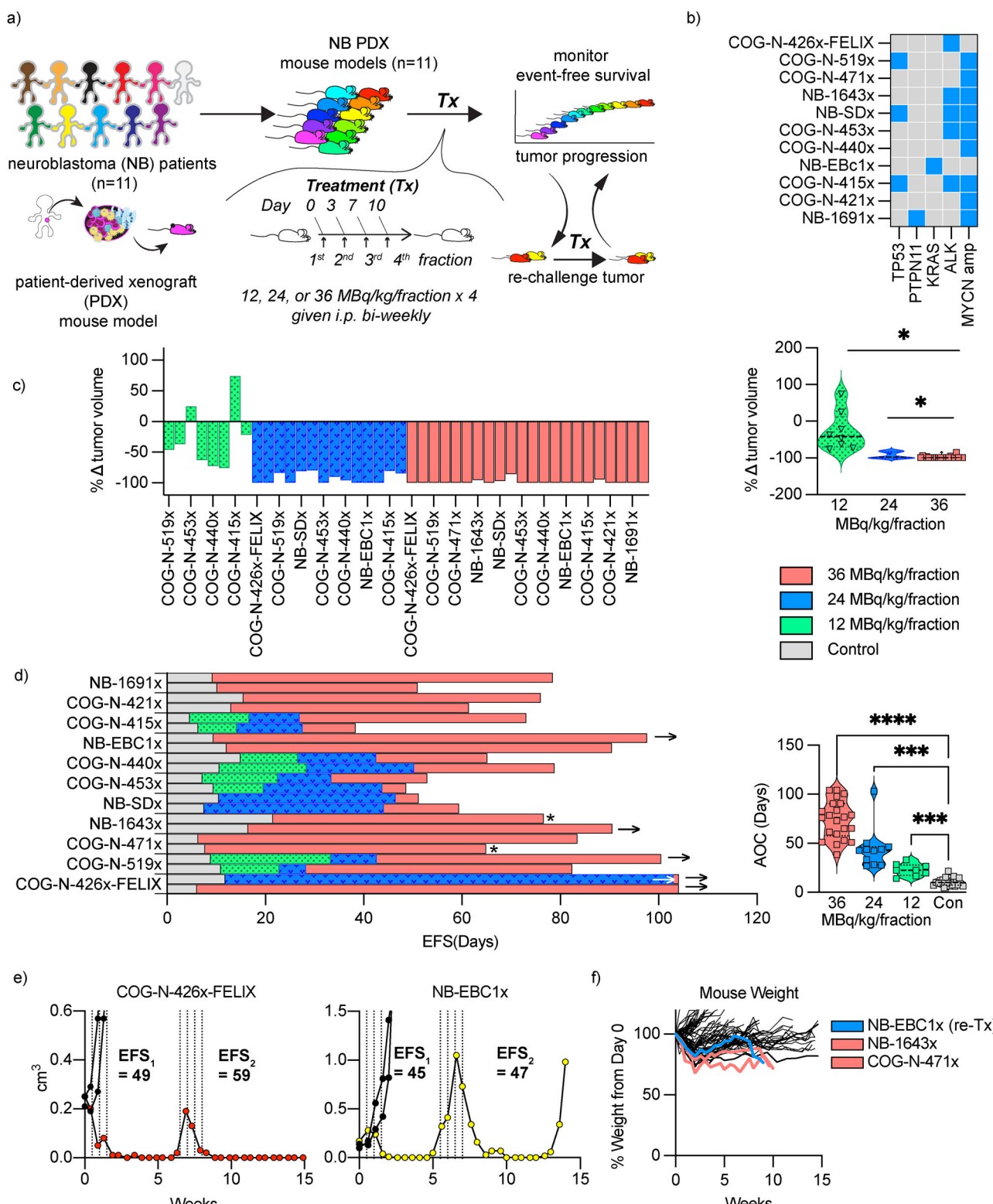

summary, [211At]PTT showed potent anti-tumor effects in a broad panel of solid tumor models at all dose levels evaluated and displayed robust activity at the MTD dose level (4 doses of 36 MBq/kg i.p twice weekly).

**Evaluation of Hematologic Toxicity.** While PARP1 expression is elevated in high-risk neuroblastoma tumors relative to the majority of normal tissues, the bone marrow compartment shows high PARP1 expression, which causes concern for on-target normal tissue toxicity from [211At]PTT[3]. Indeed, the dose-limiting toxicity of FDA-approved PARP inhibitors is hematologic toxicity[20–22]. Here, we evaluated hematologic toxicity using an immune-competent mouse model (CB57/BL6) in both male and female mice (Fig. 3a).

The MTD in CB57/BL6 mice was determined to be 48 MBq/kg/fraction administered twice weekly for a total of 4 dose fractions, which is higher than the MTD for CB57 SCID mice

**Fig. 1 Anti-tumor efficacy of [211At]PTT in high-risk neuroblastoma patient-derived xenograft models. a** Trial design for evaluation of [211At]PTT efficacy in high-risk neuroblastoma patient-derived tumor models ($n = 11$ PDX models). **b** Genetic signatures associated with PDX models. **c** Tumor response waterfall plot for all dose levels evaluated. Each bar denotes a single treated subject. Significant differences in response were found between 36 MBq/kg/fraction x 4 and lower dose levels (paired-model T-test; p-value = * <0.05). **d** Event free survival (EFS) for PDX tumors treated with [211At] PTT at 12, 24, 36 MBq/kg/fraction x 4, or saline control. Significant differences in EFS were found between control and all dose levels analyzed (paired-model ANOVA analysis; p-value = *0.013, **0.0019, ****<0.0001). **e** Tumor growth curves for PDX models treated with 24 MBq/kg/fraction x 4 and re-challenged at the same dose level at the time of tumor progression. Dotted lines indicate dose-fractionation schedule. **f** Mouse weight over time for all control and treated mice. Blue line represents NB-EBC1x ($n = 1$) mouse removed from study due to weight loss after re-challenge at 24 MBq/kg level. Salmon lines represent NB-1643x ($n = 1$) and COG-N-471x ($n = 1$) mice removed from study due to weight loss >50 days post-treatment at 36 MBq/kg dose level. * Mice removed from study due to weight loss >20% from study initiation → Mice were tumor free at end of study.

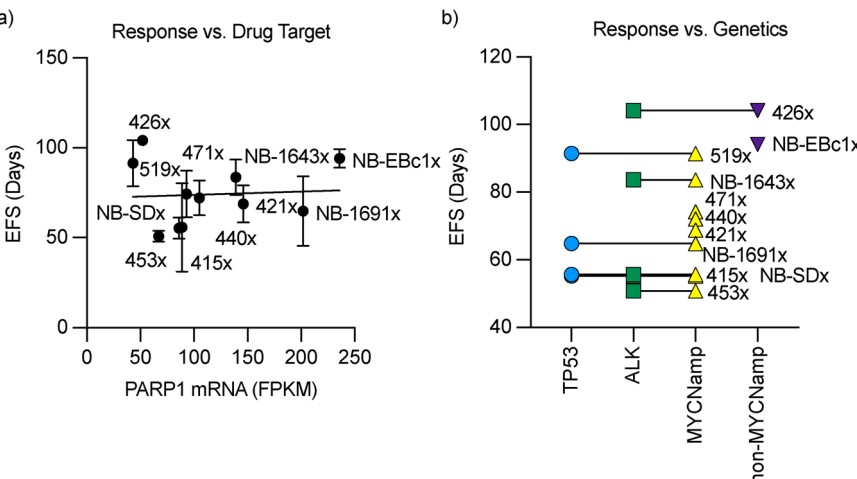

**Fig. 2 Testing for contributing factors of [211At]PTT response. a** Linear regression plot for EFS vs. *PARP1* mRNA (p value = 0.8). **b** Grouped response for genetic signatures in PDX models. Lines represent models with 1 or more genetic signatures.

(Supplementary Fig. 4a). Weight loss toxicity was dose dependent in both mouse models. Hematological toxicity at 72 hours after the final dose fraction, showed significant reductions in platelets, white blood cells, and lymphocytes at all dose levels compared to control (one-way ANOVA; p-value <0.05) (Fig. 3b). Direct hematological toxicity was evident for lymphocytes after 72 hours in groups treated with 2 or 3 dose fractions (Supplementary Fig. 4b, c), and was previously shown after only 1 dose fraction, at the 36 MBq/kg dose level[23]. This is consistent with lymphocytes having the highest PARP1 expression in the blood compartment[24]. Lymphocyte time to recovery was prolonged with each additional dose fraction although, full recovery was evident at 4 weeks post-treatment for peripheral blood counts (Supplementary Fig. 4b, c). Neutrophils, which have the lowest PARP1 expression, were largely unaffected by treatment with [211At]PTT, and have been previously shown to increase after single-dose treatment[23,24]. Platelets were reduced at all dose levels evaluated (Fig. 3b), however 3 or less dose fractions at the 36 MBq/kg dose level did not result in thrombocytopenia (Supplementary Fig. 4b, c).

In contrast to direct hematological toxicity, marrow toxicity assessed by colony formation of early marrow progenitor cells (granulocyte-erythrocyte-megakaryocyte-monocyte or GEMM) was reduced only at the highest dose level evaluated (60 MBq/kg) (Fig. 3c). Granulocyte and megakaryocyte progenitors were more sensitive to [211At]PTT and showed reduction in colony formation at all dose levels evaluated (Fig. 3c). Marrow recovery after 3 dose fractions at the 36 MBq/kg dose level stabilized by 2 weeks for GEMM and GM progenitor cells with full recovery of colony formation at 4 weeks post treatment (Fig. 3d).

The hematological and marrow toxicity were dose-dependent and showed direct effects to each compartment with maximal effects observed 72 hours post-treatment (Fig. 3c, d, Supplementary

Fig. 4b, c). To better understand the total marrow dose we performed a biodistribution study in CB57/BL6 healthy mice and isolated marrow to accurately measure radiation dose (Supplementary Fig. 4d, e, and Supplementary Table 4). The marrow dose to a 0.02 kg mouse was approximately 2.5 Gy per dose fraction at 36 MBq/kg, however, early GEMM progenitor cells appeared resilient to [211At]PTT. In summary, the toxicity profile of [211At]PTT was driven by PARP1 expression in normal tissues and was predictable, however; we observed a differential sensitivity between normal and tumor tissue that likely afforded a therapeutic window for [211At]PTT treatment.

**Biodistribution and Tumor Dosimetry.** Biodistribution of [211At]PTT was evaluated in an NB-EBC1x tumor-bearing mouse model ($n = 3$/time point) and tumor dosimetry was estimated to be 6 cGy/MBq/kg (Supplementary Table 5). Specifically, in the NB-EBC1x PDX model this corresponds to an absorbed dose of 2.15 Gy to tumor per 36 MBq/kg dose and a cumulative dose of 8.6 Gy after 4 dose fractions. In vivo deastatination of [211At]PTT was evaluated by co-administering potassium iodide with [211At]PTT ($n = 2$) and comparing distribution of unconjugated (free) astatine-211 ($n = 2$). We observed significant differences in thyroid uptake at 1 hour between [211At]PTT and [211At]PTT co-administered with SSKI (Supplementary Fig. 5a: 2-way ANOVA p value 0.0219), however, no significant differences were observed in tumor (Supplementary Fig. 5a: 2-way ANOVA p value 0.44). The biodistribution of free astatine-211 showed significantly higher uptake in thyroid and lower uptake in tumor at 1 hour compared to [211At]PTT (Supplementary Fig. 5a: 2-way ANOVA, thyroid; p-value 0.0004, and tumor; p value 0.0088). Results were corroborated by ex-vivo autoradiography that showed tumor

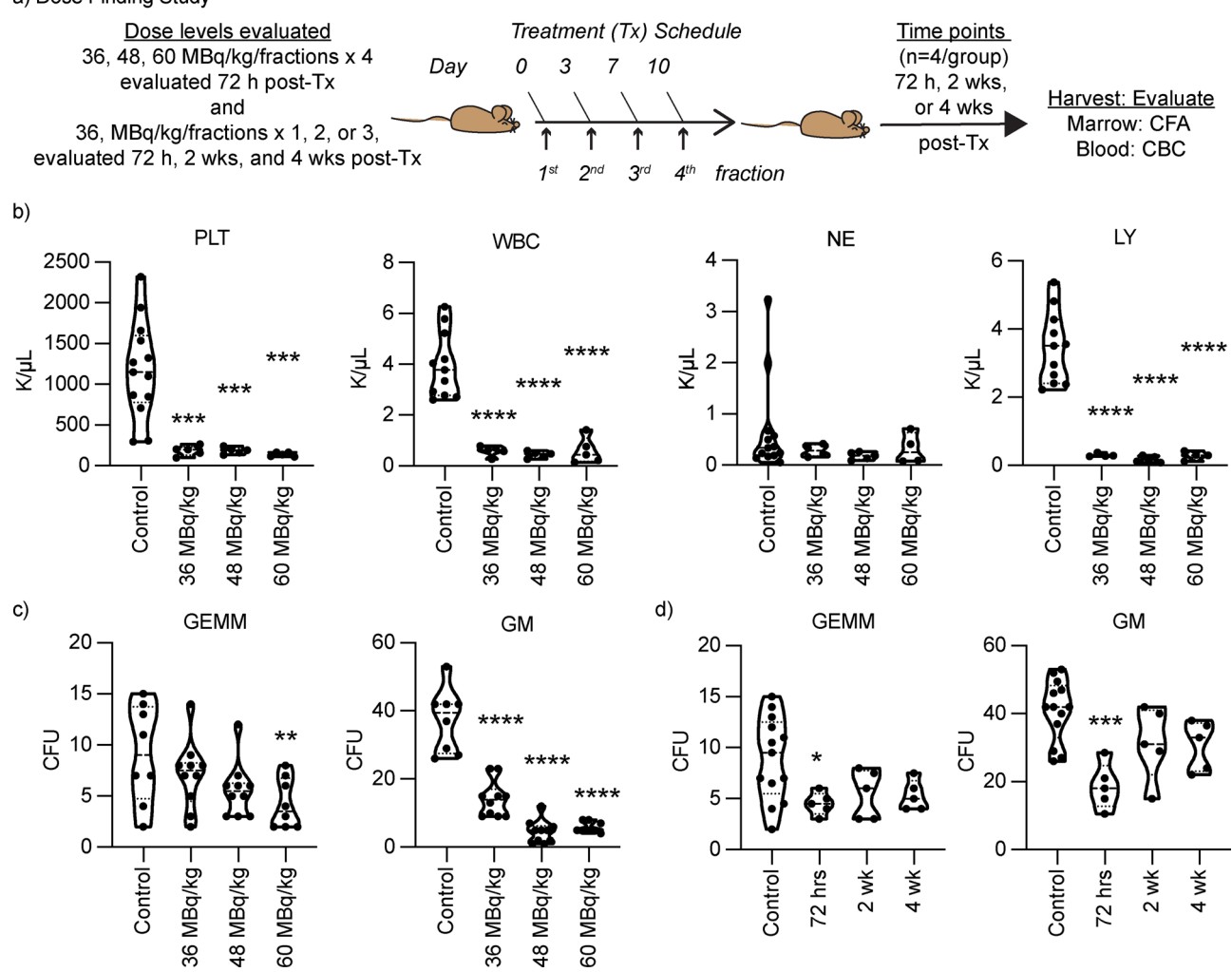

**Fig. 3 Safety profile of [²¹¹At]PTT. a** Tolerability studies to determine hematological and marrow toxicity of [²¹¹At]PTT in CB57BL6 mice. **b** Complete blood counts and **c** marrow progenitor colony formation at 72 hours post-treatment after dose escalation. Mice received either 36, 48, or 60 MBq/kg/ fraction twice weekly for a total of 4 dose fractions. **d** Marrow progenitor colony formation was assessed at 72 hours, 2 and 4 weeks post treatment for 2 dose fractions at the 36 MBq/kg/fraction dose level. Statistical analysis was performed by ordinary one-way ANOVA comparison between the mean of control and test groups. p-value denoted as * <0.05, ** <0.01, ***<0.001, ****<0.0001. PLT platelets, WBC white blood cells, LY lymphoctyes, NE neutrrophils, GEMM-early progenitor cell, GM granulocyte and macrophage progenitor cells.

uptake was greater for both, [²¹¹At]PTT and [²¹¹At]PTT + SSKI, compared with unconjugated astatine-211. Ex-vivo auto-radiography also demonstrated high tumor penetrance for [²¹¹At] PTT (Supplementary Fig. 5b). In summary, absorbed tumor dose for NB-EBC1x tumors was 2.15 Gy/dose fraction at 36 MBq/kg dose level and partial deastatination of [²¹¹At]PTT contributed to off-target accumulation of astatine-211 in thyroid but not tumor.

## Discussion
In this study we demonstrated the potent anti-tumor activity of [²¹¹At]PTT against high-risk neuroblastoma. Through a pre-clinical trial approach we evaluated [²¹¹At]PTT response in 11 neuro-blastoma PDXs that represented 11 individual patients. We observed remarkable responses in all tumor models. Inter-tumor response and drug target variability were evident, which closely resembled the reality of cancer heterogeneity in clinical trials. Variability of drug target expression across models presented a tangible explanation for differential response rates, however, no associations were found between EFS and PARP1 mRNA expression. Although PARP1 protein expression was not directly eval-uated, PARP1 mRNA has been shown to be highly correlated to

PARP1 protein expression justifying our approach[25]. Most notably, the two lowest PARP1-expressing tumor types (COG-N-519x and COG-N-426x-felix) demonstrated robust responses in vivo. This was a fascinating result given that the patient that COG-N-426x-felix was derived from was heavily pre-treated before the PDX model was created and received multiple rounds of chemotherapy, immunotherapy, radiation therapy, even [¹³¹I]MIBG radio-pharmaceutical therapy[17]. It is unclear why [²¹¹At]PTT was so effective in these models. Radiopharmaceutical therapy is often indicated in patients with tumors that express the drug target above a pre-specified threshold[26]. Based on our results in this study, we suggest caution to this approach as it may de-select patients who could respond to therapy. Radiation-induced cell damage is the mechanism of action of radiopharmaceutical therapeutics, therefore the drug target is a bystander and unlikely to infer biological out-comes. Indeed, radiopharmaceuticals often occupy less than 1% of their drug target, which is why they are referred to as "tracers".

One possible interpretation for the observed effect of drug target-independent activity could be due to host response and the tumor microenvironment. We know that [²¹¹At]PTT is toxic to lymphocytes and granulocyte/monocyte progenitor cells, therefore

reduction in host immune cells might also play a role in weakening tumors through in-direct effects. The administration of [211At]PTT also caused neutrophilia, which resembles acute stress reaction[23]. While SCID mice lack CD8 + cytotoxic t-cells, natural killer cells are still present, and combined with other phagocytic immune cells [211At]PTT could elicit an innate immune response. Immune involvement in this study is purely speculative and requires further investigation, however; preliminary evidence for synergy between [211At]PTT and anti-PD1 immune checkpoint inhibitor has been demonstrated in a syngeneic mouse model of glioblastoma[23]. Understanding alpha-particle-induced inflammation and bystander effects will be crucial for clinical implementation.

From a safety perspective [211At]PTT was tolerated, although steep dose-response effects in vivo demands a cautious approach to clinical translation. The highest absorbed doses above tumor were thyroid and marrow which are likely a mixed result of off-target accumulation of free astatine and on-target accumulation of [211At]PTT in thyroid. Marrow effects are discussed in detail later in this section. Biodistribution studies showed in vivo deastatination of [211At]PTT occurs with minimal impact to tumor uptake at 1 h. Thyroid uptake was partially blocked at 1 h with co-administration of potassium iodide. We suspect some thyroid uptake was due to [211At]PTT as previously we have shown that [18F]FTT and iodinated analogue ([125I]KX1) also exhibit thyroid uptake around 4% ID/g[8,12]. The impact of partial deastatination of [211At]PTT is likely insignificant, as it has been shown that the MTD for a single dose of free astatine-211 is greater than 50 MBq/kg[27]. To prevent thyroid accumulation in our pre-clinical efficacy studies we co-administered potassium iodide to block thyroid uptake of free astatine-211, and this approach is standard of care for patients that receive radioligand therapy with iodine-131-*meta*-iodobenzylguanidine[4]. We conclude that in vivo deastatination is likely insignificant with regards to both efficacy and toxicity of [211At]PTT.

Dosimetry extrapolated from biodistribution of [211At]PTT in mice to a 1-year-old human phantom predicts that the kidneys would receive 0.026 Gy per MBq/kg, which translates to 0.34–0.44 mGy/MBq when scaled for an adult. From first-in-human dosimetry studies of companion diagnostic [18F]FTT we can estimate the kidney dose of [211At]PTT; however, in doing so we assume that both agents behave similarly in vivo. The estimated kidney dose for [18F]FTT in adults was 0.018–0.027 mGy/MBq (8), and using the same human biodistribution results we obtain the estimated renal absorbed dose of 0.25–0.41 mGy/MBq for [211At]PTT. This estimation is in agreement with our dose estimate of 0.34–0.44 mGy/MBq derived from the mouse model, despite the assumptions involved in extrapolation between different radiotracers and species[28]. Using the relative biological effectiveness (RBE) of 5 for alpha radiation[29], the extrapolated equivalent dose to the kidneys for [211At]PTT is approximately 2 mSv/MBq, only a small fraction of the kidney dose for the radiopharmaceutical therapeutic [225Ac]PSMA-617 (740 mSv/MBq) that has entered clinical trials for the treatment of castrate resistant metastatic prostate cancer[29]. While the large difference in renal radiation dose between [211At]PTT and [225Ac]PSMA-617 is partly explained by the longer physical half-life (10 days vs. 7.2 hours) and the higher alpha yield (four vs. one per decay) of actinium-225[30], it is also representing the established route of hepatobiliary clearance accounting for >90% of blood clearance for rucaparib (analogue PARPi) and companion diagnostic [18F]FTT[8,31]. Although renal elimination is partially observed in mice we have not observed this clinically with [18F]FTT[8]. From a safety standpoint, the low kidney dose from [211At]PTT will allow for hyper-fractionated dosing schedules such as the one used in our pre-clinical efficacy studies.

A rudimentary estimation for the first-in-human tumor dose of [211At]PTT can be derived from clinical PET imaging data with companion diagnostic [18F]FTT, which has demonstrated the maximum standardized uptake values (SUVmax) of 2–9 in ovarian cancer and 4–7 in breast cancer (14[32]). Neuroblastoma is expected to express PARP1 at levels twice that of solid tumors in adults[33], and we have demonstrated this in pre-clinical models where peak uptake with [18F]FTT was 2 to 4% ID/g in breast cancer models HCC1937 and MDA-MB-231[34], while [211At]PTT uptake was 10 to 15% ID/g for neuroblastoma models IMR-05[3] and NB-EBC1 (this work). We could estimate [211At]PTT SUVmax in neuroblastoma tumors to reach approximately 10, although in clinical settings SUVmax would be highly variable among different patients and even within a patient. Furthermore, we are unable to accurately estimate absorbed tumor dose from the prior [18F]FTT study as the imaging window of up to 200 minutes from injection failed to reveal a washout pattern[14]. By using the biological half-life seen in mice in the present study (5.1 hr) for [211At]PTT and then applying the SUVmax of 10, we can make a rough estimate of the absorbed tumor dose in a 1-year-old patient to be 0.17 Gy per MBq/kg. Given that this is an overestimation due to use of SUVmax rather than SUVmean, the result is comparable to the average tumor dose of 0.060 Gy per MBq/kg in a 1-year-old phantom derived from our NB-EBC1 tumor-bearing mouse model. A major limitation of the current pre-clinical trial design is that late toxicities of [211At]PTT were not directly assessed in a healthy mouse model, however, this does not limit clinical translation to First-In-Human phase 1 and phase 2 clinical trials under FDA guidelines that recommend single species long term toxicity studies at the time of marketing[35].

On target toxicities to normal tissues were predictable and are consistent with clinical PARP inhibitors that also bind PARP1 in marrow and blood cells[20–22]. The use of a fractionated dosing regimen helped mitigate normal tissue toxicity while maximizing anti-tumor efficacy and suggested peak concentrations drive toxicity. This notion is supported from early work that demonstrated 1,480 kBq administered as a single dose was lethal, but separated into 4 dose fractions given twice weekly was tolerated[3]. Therefore, normal tissue toxicity could be avoided by reducing peak activity and tumor burden could be reduced sequentially at nearly the same pace for tumor doubling time. Normal tissues also appeared less sensitive to [211At]PTT. This effect was most evident for early marrow progenitor cells (GEMM) which displayed remarkable resiliency. Another aspect that could affect [211At]PTT toxicity to normal tissues is the heterogenous expression of PARP1 that exhibits more cellular than tissue specificity[36]. In general, early progenitor cells, immature cells, and highly proliferative cells express the highest level of PARP1 but only exist as sub-populations in normal tissues[36]. For example, lung bronchiole cuboidal cells have low PARP1 expression, while bronchial ciliated columnar cells show moderate expression, and Type 1 pneumocytes have either high expression or no expression at all[36]. The heterogenous expression of PARP1 in normal tissues complicates general whole organ dosimetry approaches, especially for [211At]PTT as an alpha-emitter with exquisite single-cell toxicity. Future studies that address late-tissue toxicity must consider sub-populations within tissues to understand the risks associated with [211At]PTT.

In summary we have evaluated [211At]PTT for the treatment of high-risk neuroblastoma and the results of our study demonstrate the clinical viability of [211At]PTT, however, the safety profile will require [211At]PTT investigation in adult neuroblastoma patients before evaluation in pediatric neuroblastoma patients. Furthermore, this study broadly demonstrates the utility of small molecule alpha-emitting radiopharmaceutical therapeutics that utilize 211At, and this example sets precedence for the future development of similar agents in this drug class.

## Methods

**Radiopharmaceuticals**. Astatine-211 was produced at the University of Pennsylvania Cyclotron Facility and [211At]PTT was prepared through electrophilic aromatic destannylation of a tin precursor as previously described[7]. Final product was measured on a Capintec (Florham Park, NJ) dose calibrator and diluted to the desired concentration with appropriate medium.

**Patient-derived xenograft tumor models**. Eleven neuroblastoma patient-derived xenograft models were evaluated in this study. Models were provided by the Children's Hospital of Philadelphia. Informed consent was obtained from each research subject or legal guardian under protocol approved by the Institutional Review Board at the Children's Hospital of Philadelphia. Tumor models have been deidentified in the presently reported study.

**Animal studies approval**. All animal studies were approved by the Children's Hospital of Philadelphia Institutional Animal Care and Use Committee. Animal experiments were conducted in an accredited pathogen-free facility at the Department of Veterinary Research (DVR), Children's Hospital of Philadelphia. PDX models were all characterized with DNA and RNA sequencing[17]. All blood and marrow toxicity studies performed in animals were done under a protocol approved by the Institutional Animal Use and Care Committee at the University of Pennsylvania.

**Pre-clinical efficacy studies**. Neuroblastoma patient-derived xenograft models were prepared as previously described and [211At]PTT efficacy was tested in 11 PDX tumor models ($n = 2$/arm)[19]. The use of 2 mice per arm was based on the previously reported statistical analysis[19]. PDX tumors were implanted into the right flank of 5 to 6-weeks old female mice weighing about 20–25 grams. Animals bearing engrafted tumors between $0.2\,cm^3$ to $0.3\,cm^3$ were randomly assigned into 2 cohorts of $n = 2$ mice per group; vehicle control or [211At]PTT treatment. Intraperitoneal injections (IP) of SSKI were given for thyroid protection to prevent the accumulation of astatine-211 liberated from [211At]PTT in vivo. The maximum tolerated dose (MTD) of 36 MBq/kg of [211At]PTT was given in 4 fractionated doses intraperitoneally on day 0, 3, 7, and 10. Two additional dose levels of 12, and 24 MBq/kg/fraction were assessed in a subset of models and were administered as stated above. In two models treated at the 24 MBq/kg/fraction dose level we re-challenged tumor response at the time of tumor progression with an additional 4 dose fractions at 24 MBq/kg/fraction given twice weekly. Doses were prepared in 500 μL total volume. An equal volume of saline was used for the vehicle group. Tumors were measured using a digital caliper at the initiation of the study and two times per week during the treatment period. Tumor volumes were calculated as volume = $((diameter^1/2 + diameter^2/2)^3 * 0.5236)/1000$. Weights were obtained twice weekly and mice were monitored daily for signs of toxicity. Mice were euthanized when tumor volumes reached/exceeded $2\,cm^3$ or if an animal displayed signs of clinical toxicity, including excessive weight loss (more than 20% initial weight), dehydration, or respiratory distress. Tumor response was assessed by best response defined as greatest reduction in tumor volume following the last dose and is presented as percent change in tumor volume. Event-free survival was determined by area over the curve method that calculates the time to tumor event, which was defined as tumors reaching or exceeding a size $\geq 2\,cm^3$.

**Evaluation of [211At]PTT hematological and marrow toxicity**. Toxicity studies were done in nontumor bearing CB-17 SCID mice (Taconic Biosciences). Both male and female mice were used to evaluate toxicity with [211At]PTT. Each arm had n = 10 mice (5 male, 5 female mice). A broad range of doses (12, 24, 36, and 48 MBq/kg/fraction) were used to determine the MTD of [211At]PTT. All mice were administered intraperitoneal injections (IP) of a saturated solution of potassium iodide (SSKI), 100 mg/kg, in a volume of 100 microliter/10 gram body weight, on the day prior to, day of, and a day after their respective [211At]PTT injections for thyroid protection. SSKI injections started on the day of enrollment (Day 0) and the first dose fraction of [211At]PTT was administered on Day 1 followed subsequently by three dose fractions delivered 72 hours apart. [211At]PTT doses were prepared in 500 μL total volume. The vehicle group received an equal volume of saline (placebo). Weights were obtained twice weekly and mice were monitored daily for signs of toxicity. Mice were euthanized according to IACUC protocol when body weight dropped below 20% of baseline weight at enrollment or if an animal displayed signs of clinical toxicity, including excessive dehydration, respiratory distress. On target hematological and marrow toxicity which was evaluated by complete blood counts and marrow colony formation assay. An additional study was performed under identical conditions to evaluate additive dose effects. We randomly assigned mice into four groups that received bilateral intraperitoneal injections of either saline or [211At]PTT, at doses corresponding to 12, 24, and 36 MBq/kg of activity in saline, with a total injection volume of 500 μL. Each treatment group was composed of equal numbers of male and female mice to rule out gender-specific effects ($n = 4$ per treatment group per time point). Mice were followed for signs of acute toxicity for 2 weeks and 4 weeks before euthanasia. An additional group ($n = 4$) treated at 36 MBq/kg was euthanized after 72 h to assess early signs of toxicity by colony formation assays. This time course was chosen to best capture the nadir of bone marrow suppression. After euthanasia, blood was collected and femurs were dissected and processed for histopathology and bone marrow evaluation.

Animals were monitored weekly for weight and daily for signs of distress until the time of euthanasia. To assess hematological toxicities of [211At]PTT, we performed complete blood count analysis at the time of euthanasia using a HEMAVET 950 FS blood analyzer *(The Americas Drew Scientific Inc., Oxford CT)*. Mice were anesthetized with mixed inhalation isoflurane/oxygen under a nosecone. Terminal cardiac puncture was performed and peripheral blood was transferred to $K_2$EDTA BD Microtainer tubes (BD, San Antonio, TX). Sample volume injected was 20 μL and was pre-specified using the mouse protocol on the instrument.

After euthanasia, mouse hind limbs were dissected as previously described for bone and marrow isolation. The other hind limb was processed for bone marrow via centrifugation. Bone marrow pellets were resuspended in 1 X RBC Lysis Buffer (BioLegend, San Diego, CA) and rocked on a tilt-table for 5 minutes. Bone marrow cell pellet was isolated by centrifugation at 13,000 RPM on a tabletop centrifuge for 1 minute. Cells were resuspended in IMDM with 2% FBS (Gibco, ThermoFisher Scientific, Grand Island, New York) and counted on a Countess II Automated Cell Counter (ThermoFisher Scientific, Grand Island, NY).

Colony formation assays were performed per manufacturer's protocol. In brief, $1.8 \times 10^5$ bone marrow cells suspended in 300 μL IMDM with 2% FBS were combined with 3 mL of Mouse Methylcellulose Complete Media (R&D Systems Inc, Minneapolis, MN). Gel mixture was syringe-pipetted in duplicate to lids of 6 cm culture dishes and housed with a 6 cm sterile water dish within a 25 cm culture dish. Cultures were incubated at 37 °C for 8 days prior to analysis on a microscope. Colonies were counted by one individual per manufacturer's recommended protocol. Of note, at least 30 cells were required to call a colony positive.

Biodistribution of [211At]PTT was performed in a healthy mouse model CB57/BL6 using female mice. Using female mice was justified due to the lack of sex-related differences observed from initial dose-finding studies. Mice were anesthetized by isoflurane administered by nose cone and 185 kBq of [211At]PTT was administered by intraperitoneal injection. At various time points mice were euthanized and tissues were harvested, weighed, and assayed for radioactivity by gamma counter. Decay corrected data is presented as percent injected dose per gram of tissue (%ID/g). Human dosimetry was calculated using OLINDA/EXM v1.1 software by fitting time activity curves for biodistribution data for a 3-year-old and 5-year-old phantom model, which represents the typical age of high-risk neuroblastoma patients.

**Biodistribution and tumor dosimetry**. Biodistribution of [211At]PTT was performed in a NB-EBC1x tumor bearing mouse model (female SCID CB17 mice) alongside [211At]PTT + SSKI and unconjugated astatine-211. Tumor uptake for [211At]PTT was evaluated at 0.5, 1, 4, and 24 hours and at 1 hour for [211At]PTT + SSKI and unconjugated astatine-211. Mice were anesthetized by isoflurane administered by nose cone and 185 kBq of [211At]PTT ($n = 3$), [211At]PTT + SSKI ($n = 2$), or unconjugated astatine-211 ($n = 2$) was administered by bilateral intraperitoneal injection (250 μL on each side). At various time points mice were euthanized and tissues were harvested, weighed, and assayed for radioactivity by gamma counter. Decay corrected data presented as percent injected dose per gram of tissue (%ID/g). Human dosimetry was calculated using OLINDA/EXM software by fitting time activity curves for biodistribution data. Tissue autoradiography was performed by freezing excised tumors at 1 h postinjection of each respective treatment. One tumor was collected for each treatment group. Tumors were cryosectioned (Leica Cryotome) at a thickness of 20 μm. Each section was collected with a 200 μm gap and a total of three sections were collected for each tumor. Sections were then exposed to a phosphorfilm for 60 h and films were read on a Cyclone Plus phosphorimager (Perkin Elmer). Experimental conditions have been included in the Supplementary Methods.

**Statistics and reproducibility**. Animal studies were performed in female mice due to the lack of sex-related differences for [211At]PTT tolerability from initial dose response studies performed in female and male mice. Female mice were chosen due to their underlying behavior towards other study subjects that reduced nonstudy-related causes for decreased wellness. Pre-clinical efficacy studies were performed through "single mouse testing" to more closely recapitulate a clinical trial[19]. This design was used to test a large effect size and have sufficient diversity in xenograft models to explore broad antitumor effectiveness of [211At]PTT. Number size for treatment groups was determined by model tumor growth reproducibility, which was previously shown to have less than 5% normal variation with at least two animals[19]. Statistical tests for tumor response and event-free survival (EFS) were performed as a grouped analysis that included all tumor models treated at each respective dose level ($n = 2$ per model per dose level). Statistical differences between treatment groups for tumor response was performed by paired-model T-test that compared reduction in tumor volume for the 36 MBq/kg dose level vs. 24 MBq/kg or 12 MBq/kg. Event free survival was measured by area over the curve method to quantify time to event, which was set as tumor growth exceeding $2\,cm^3$[19]. Differences in EFS were tested across all treated models for each respective dose level, and against untreated controls, by paired-model ANOVA analysis. Associations of drug target expression and response were performed by linear regression analysis of EFS and *PARP1* mRNA expression. Genetic signatures

were visually presented as a function of EFS and no statistical tests were applied. Blood and marrow toxicity studies were performed with $n = 5$ sample sizes and powered to detect >20% differences between untreated controls and treated animals. Results were tested for significance by one-way ANOVA comparison from untreated controls. Marrow colony formation assays were performed in replicates of 2 for each mouse subject, dependent on marrow isolation, and all replicates are included in final analyses. Statistical analysis was performed by ordinary one-way ANOVA comparison between the mean of control and test groups. Biodistribution studies were performed with $n = 3$ mice per time-point. Human dosimetry was calculated using OLINDA/EXM v1.1 software by fitting time activity curves for biodistribution data in a healthy mouse model. Tumor dosimetry was calculated by OLINDA/EXM v1.1 by fitting time activity curves for NB-EBC1 tumors from biodistribution data. All statistical analyses were performed in GraphPad Prism version 9.3.1 unless indicated otherwise.

**Reporting summary**. Further information on research design is available in the Nature Portfolio Reporting Summary linked to this article.

## Data availability

Authors of this manuscript will do their best to accommodate requests for resources or data used or generated. Raw data used to generate figures in the main text have been included as a separate file titled "Supplementary Data 1". All resources and data will be made available upon request, within reason, to corresponding author Dr. John M. Maris at maris@chop.edu.

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

## Acknowledgements

Astatine-211 isotope production was supported partly by DOE-NP Isotope Program award DE-SC0021066 (M Makvandi). All aspects of this work were supported fully or in part by NIH/NCI awards R01CA219006 (DA Pryma), R35CA220500 (JM Maris), and the Giulio D'Angio Endowment (JM Maris). We would like to acknowledge the cyclotron facility manager HS Lee, and cyclotron engineers D Schaub and L Toto at the University of Pennsylvania Cyclotron Facility for supplying astatine-211.

## Author contributions

[1,3,4,5]M.M., [2]M.S., [2]P.M., [2,3]H.L., [2,3]S.B.G., [3]K.P., [2]D.G., [1,2,3]J.P., [2,3]D.M., [2]A.R., [2]H.D., [2]M.Z., [2]T.T., [2]K.X., [2]J.YL., [2]C.H., [2,3,5]A.F., [1,4]V.B., [1,4]S.D.C., [5]D.J.P. Jr, [5]R.H.M., [1,3,4,5]D.A.P., and [1,3,4,5]J.M.M. [1]Methodology [2]Conducting Experiments [3]Data visualization [4]Writing, reviewing, and editing Manuscript [5]Resources and Funding.

## Competing interests

JM Maris is a paid consultant for Jubilent Radiopharma and Illumina Radiopharmaceuticals. DA Pryma discloses research grants from Siemens AG, 511 Pharma, and Progenics Pharmaceuticals Inc; research consultant positions with 511 Pharma, Progenics Pharmaceuticals Inc., Ipsen, and Actinium Pharmaceuticals Inc; and Clinical Trial Funding from Nordic Nanovector ASA. RH Mach, DA Pryma, and M Makvandi

are listed as inventors on the USA Patent Number PCT/US2018/034398 held by the University of Pennsylvania that describes radiotherapeutic agent [211At]PTT. RH Mach is co-founder and scientific advisor for Trevarx biomedical which has licensed exclusive rights for [211At]PTT. All other authors are free from any competing interests.
