## [Peer Review File · Communications Biology]

Reviewers' comments:

Reviewer #1 (Remarks to the Author):

This paper by Makvandi et al. reports promising results from a pre-clinical trial on alpha-particle-based radiopharmaceutical therapy using a small molecule PARP1-inhibitor to target tumor cells. The tumor model, consists of 11 different murine PDX models generated from neuroblastoma tumor tissue with varying clinical and genetic characteristics. The results are highly relevant for the development of novel therapy for high-risk neuroblastoma, given the relatively high radiation sensitivity of neuroblastoma tumors and the potential of this approach to deliver a lethal radiation dose to metastatic sites. The methodology and statistical analyses appear to be adequate and results are presented clearly in text and in the supplied figures.

There are a few claims made by the authors that require clarification and / or discussion:

1) In the introduction, the authors claim that "Unlike radiometal alphaemitter complexes, such as chelated actinium-225 that has poor cellular permeability, ^{211}At is a radiohalogen that can be covalently incorporated into small molecule drugs that provides a unique platform for targeting intracellular proteins." This statement seems to be based on theoretical assumptions and should be supported by references to clarify whether this claimed benefit has been shown to be relevant in a clinical or pre-clinical setting.

2) In the results section, "Evaluation of hematologic toxicity" is thoroughly presented. Here the authors claim that "While PARP1 expression is elevated in high-risk neuroblastoma tumors relative to the majority of normal tissues, the bone marrow compartment shows high PARP1 expression which causes concern for on-target normal tissue toxicity from ^{211}At PTT (3)". PARP1 is however, apart from the bone marrow compartment, also widely expressed in many other normal tissues. Results on dosimetry are presented in the Supplemental Information, Table 4, displaying theoretical estimated doses in humans and showing high estimated doses for the thyroid and even more importantly, relatively high doses in radiosensitive organs, such as the lungs and kidneys. Radiation damage to the kidneys and lungs typically causes late toxicity, which would not become evident with the current pre-clinical trial design. The risk of such toxicity is, however, highly relevant for the translation of these promising results to clinical trials in humans. While stated that "Evaluation of gross tissue toxicity was beyond the scope of this study and will be investigated in the future." the discussion section would benefit from including the results of Table 4 in the Supplemental Information and discussing the consequences and the relevance of potential late toxicity to organs at risk.

Reviewer #2 (Remarks to the Author):

This interesting paper describes the efficacy and toxicity of the PARP inhibitor parthanatine, radiolabeled with the alpha particle emitter astatine-211, in 11 patient derived neuroblastoma murine xenografts. As a treatment administered in four fractions over ten days, this radiopharmaceutical appears to be very effective with limited and manageable toxicity. The work is original, scientifically sound, and convincing.

It is of interest that the efficacy was not related to the level of PARP expression. Hopefully a clinical evaluation may follow, subject to all the usual research governance requirements for a novel radiopharmaceutical, as patients with high-risk neuroblastoma have a poor outcome, and innovative therapeutic approaches are required.

This manuscript makes no mention of any imaging studies using the same vector labeled with a different radionuclide selected for imaging purposes. I think for safe clinical use a theragnostic partner compound would be considered necessary to demonstrate localisation in all tumour deposits, and ideally not in critical normal tissues or organs. The manuscript could usefully be strengthened by discussion of this.

Mark Gaze

Reviewer #3 (Remarks to the Author):

This paper by Makvandi et al. describes an alpha-emitting radiopharmaceutical, based on the PARP inhibitor rucaparib. The manuscript contains in vivo evaluation in a variety of NB PDX models, with evaluation of hematological toxicity.

PTT structure not defined in this paper. Please make it more clear that it is the same as 211At-MM4. How similar is it to rucaparib? What are the tumor targeting kinetics, slow target off-rate, and deep tumor penetration of PTT?

SCID mice are known to be hypersensitive to ionising radiation (Biedermann et al. PNAS 1991). Why were they chosen as hosts for PDX to evaluate an alpha particle emitter?

Please justify more the choice for NB as a tumor for TAT? If 131I-MIBG is already successful, an alternative may not be needed? How much better is 211At-PTT compared to 131I-MIBI? 'Given the impressive single agent activity of beta-emitting radiopharmaceutical therapeutic – iodine-131-metaiodobenzylguanidine, we sought to develop a PARP1 targeted radiopharmaceutical therapeutic that uses an alpha-emitting radionuclide to overcome the biophysical limitations of beta-therapy'. Given potential off-target toxicity, would one consider alpha particle therapy in pediatric patients? 'We seek to deliver alpha-radiation directly to cancer cell nuclei': Given the 70 um range of alpha particles, why direct the radiation to cancer cell nuclei?

How variable was PARP expression among each PDX tumor?

211At-MM4 was previously shown to be effective in PARP-expressing cancer models. What is added value here? Only xenograft -> PDX .

Is an MTD relevant for TAT therapy? It seems the lower dose regimen may have a better therapeutic index.

What was the uptake of 211At-MM4 in NB tumors? PARP expression? Dosimetry?

Given the late-stage weight loss in some animals, was the cause investigated? Were there late toxicities? Was histology or other damage in normal tissue evaluated?

Fig 1: Are any control animals included, to probe tumour growth in case of no treatment (fig 1c)? It is impossible to gauge how large tumors are at the start of therapy? EFS for control animals seems very short?

SI Fig 2: were only 2 animals per group treated with 211At per PDX? Although therapeutic affect is observed across the PDXs, does this allow statistical analysis? What is the relevance of the different PDX models, if no conclusion can be drawn regarding the individual PDX models? The publication that is referenced to justify n=2, regards drug screening, not multiple model screening. The relevance of this reference, and the choice of n=2 requires additional justification.

Tumor rechallenge in only 2 animals is anecdotal at best. How much conclusion can be drawn from 2 animals?

It seems mouse weight was considerably affected by 211At-PTT (SI Fig 2C). Was mouse weight loss dependent on administered dose? Some animals seem to lose >20% of their body weight? Was this corrected for tumor size?

How is thyroid uptake explained? Is this an issue (given the radiation dose). What is the radiation dose to the tumor?

Is there de-astatination? Is tumor uptake specific (blockable)?

Point-by-point response

NOTE: Our response to reviewer concerns has been appended (shown in purple) to reviewer comments (shown in black).

Reviewers' comments:

Reviewer #1 (Remarks to the Author):

This paper by Makvandi et al. reports promising results from a pre-clinical trial on alpha-particle-based radiopharmaceutical therapy using a small molecule PARP1-inhibitor to target tumor cells. The tumor model, consists of 11 different murine PDX models generated from neuroblastoma tumor tissue with varying clinical and genetic characteristics. The results are highly relevant for the development of novel therapy for high-risk neuroblastoma, given the relatively high radiation sensitivity of neuroblastoma tumors and the potential of this approach to deliver a lethal radiation dose to metastatic sites. The methodology and statistical analyses appear to be adequate and results are presented clearly in text and in the supplied figures.

There are a few claims made by the authors that require clarification and / or discussion:

1) In the introduction, the authors claim that "Unlike radiometal alphaemitter complexes, such as chelated actinium-225 that has poor cellular permeability, ^{211}At is a radiohalogen that can be covalently incorporated into small molecule drugs that provides a unique platform for targeting intracellular proteins." This statement seems to be based on theoretical assumptions and should be supported by references to clarify whether this claimed benefit has been shown to be relevant in a clinical or pre-clinical setting.

The introduction has been modified to address this concern citing references that support concepts discussed.

"Conjugation chemistry of radiometals, such as ^{225}Ac , requires metal chelation and is not suitable for organic small molecule targeting vectors that must diffuse across cell membranes. In fact, only 0.9% of ^{225}Ac chelated with dodecane tetracetic acid (^{225}Ac -DOTA) was shown to associate with cells and did not appear to penetrate cells {Zhu, 2017 #29}. Unlike radiometal alpha-emitter complexes, ^{211}At is a radiohalogen that can be covalently incorporated into small molecule drugs that provides a unique platform for alpha-therapeutics targeted to intracellular proteins. Furthermore, sufficient tumor diffusion of alpha-emitting therapeutics has been shown to be a critical component for optimal tumor response {Howe, 2021 #30}, and organic small molecule PARPi exhibit large volume of distribution that suggests rapid transfer of drug from circulation to peripheral tissues. In this study we seek to deliver alpha-radiation directly to cancer cell nuclei by targeting PARP1."

2) In the results section, "Evaluation of hematologic toxicity" is thoroughly presented. Here the authors claim that "While PARP1 expression is elevated in high-risk neuroblastoma tumors relative to the majority of normal tissues, the bone marrow compartment shows high PARP1 expression which causes concern for on-target normal tissue toxicity from [^{211}At]PTT (3)". PARP1 is however, apart from the bone marrow compartment, also widely expressed in many other normal tissues. Results on dosimetry are presented in the Supplemental Information, Table 4, displaying theoretical estimated doses in

humans and showing high estimated doses for the thyroid and even more importantly, relatively high doses in radiosensitive organs, such as the lungs and kidneys. Radiation damage to the kidneys and lungs typically causes late toxicity, which would not become evident with the current pre-clinical trial design. The risk of such toxicity is, however, highly relevant for the translation of these promising results to clinical trials in humans. While stated that "Evaluation of gross tissue toxicity was beyond the scope of this study and will be investigated in the future."

We identified a mistake in the dosimetry table and the units for radiation dose have been adjusted accordingly. The dosimetry SI table 4 had incorrect values for units of Gy/MBq, while the graph in SI figure 3E had correct values for units of mGy/MBq. We have corrected the values of SI table 4 to match the units of Gy/MBq.

The discussion has been significantly modified to discuss safety aspects of [211At]PTT. We thank the reviewer for critically evaluating the aspects of safety in this work, which allowed us to take a closer look at these significant concerns, make corrections, and expand within the topic.

Reviewer #2 (Remarks to the Author):

This interesting paper describes the efficacy and toxicity of the PARP inhibitor parthanatine, radiolabeled with the alpha particle emitter astatine-211, in 11 patient derived neuroblastoma murine xenografts. As a treatment administered in four fractions over ten days, this radiopharmaceutical appears to be very effective with limited and manageable toxicity. The work is original, scientifically sound, and convincing.

It is of interest that the efficacy was not related to the level of PARP expression. Hopefully a clinical evaluation may follow, subject to all the usual research governance requirements for a novel radiopharmaceutical, as patients with high-risk neuroblastoma have a poor outcome, and innovative therapeutic approaches are required.

This manuscript makes no mention of any imaging studies using the same vector labeled with a different radionuclide selected for imaging purposes.

I think for safe clinical use a theragnostic partner compound would be considered necessary to demonstrate localisation in all tumour deposits, and ideally not in critical normal tissues or organs.

The manuscript could usefully be strengthened by discussion of this.

The introduction and discussion have been revised to include companion diagnostic [18F]FTT. We thank the reviewer for this incredibly helpful suggestion for strengthening the paper.

Mark Gaze

Reviewer #3 (Remarks to the Author):

This paper by Makvandi et al. describes an alpha-emitting radiopharmaceutical, based on the PARP inhibitor rucaparib. The manuscript contains in vivo evaluation in a variety of NB PDX models, with evaluation of hematological toxicity.

PTT structure not defined in this paper.

Refer to citations 3 and 7. Chemical structures of previously identified entities can be referenced without displaying the structures.

Please make it more clear that it is the same as 211At-MM4. How similar is it to rucaparib?

We have added an additional reference plainly stating that [211At]PTT is the same as [211At]MM4.

“By radiolabeling a small molecule PARP inhibitor with astatine-211 (²¹¹At), we developed the first-in-class alpha-emitting drug that targets cancer nuclei via PARP1, [²¹¹At]parthanatine ([²¹¹At]PTT) – previously referred to as [²¹¹At]MM4 (7).”

“In early proof of concept studies we demonstrated that [²¹¹At]PTT (previously referred to as [²¹¹At]MM4) was nearly 1 billion times more cytotoxic than the non-radioactive PARP inhibitor analog (KX1) (3).”

What are the tumor targeting kinetics, slow target off-rate, and deep tumor penetration of PTT?

Tumor targeting kinetics were extrapolated from clinical PET imaging studies with companion diagnostic [18F]FTT with the disclaimer “this assumes both agents behave similarly in vivo”. Furthermore, kinetic evaluation of [211At]PTT was beyond the scope of this work and is currently ongoing.

SCID mice are known to be hypersensitive to ionising radiation (Biedermann et al. PNAS 1991). Why were they chosen as hosts for PDX to evaluate an alpha particle emitter?

These mice are required for the PDX cells to grow optimally, particularly in metastatic models. While these mice are sensitive to the effects of ionizing radiation, that sensitivity does not translate to the transplanted PDX cells. Therefore, these models are still a good test of efficacy against the PDX cells and the fact that the treatment is tolerable in the radiation sensitive SCID mice lends encouragement that PTT will be tolerable when translated to human use. Indeed, we did show that the MTD in SCID mice is lower than in CB57/BL6 mice.

Please justify more the choice for NB as a tumor for TAT? If 131I-MIBG is already successful, an alternative may not be needed? How much better is 211At-PTT compared to 131I-MIBI? ‘Given the impressive single agent activity of beta-emitting radiopharmaceutical therapeutic – iodine-131-metaiodobenzylguanidine, we sought to develop a PARP1 targeted radiopharmaceutical therapeutic that uses an alpha-emitting radionuclide to overcome the biophysical limitations of beta-therapy’.

NB is a radiation sensitive cancer but despite the high response rate to MIBG it is not curative. Given the high mortality of relapsed/refractory NB there is definitely an unmet need for more effective therapies.

One of the major putative limitations in MIBG efficacy relates to microscopic residual disease which is going to be more effectively targeted with an alpha emitter than a beta.

Given potential off-target toxicity, would one consider alpha particle therapy in pediatric patients?

Because relapsed/refractory NB has such high mortality, current treatment approaches accept considerable toxicity. Based on our preclinical testing, we are confident that the toxicity from PTT is expected to be acceptable for translation to human testing.

'We seek to deliver alpha-radiation directly to cancer cell nuclei': Given the 70 um range of alpha particles, why direct the radiation to cancer cell nuclei?

While the range of an alpha particle spans more than one cell, the probability of directly hitting the DNA and causing a double strand break (which is plausible given the dense ionizations caused by alpha particles) is exponentially higher when the alpha originates in the nucleus (and even higher when bound to the DNA at the time of emission) compared to an alpha originating from the cytoplasm or exterior to the cell. Therefore, there is additional potential therapeutic ratio from a nuclear target.

How variable was PARP expression among each PDX tumor?

PARP1 expression was not evaluated however PARP1 mRNA expression is listed in the results section and SI table 3. PARP1 mRNA expression has a high correlation to PARP1 protein expression so we felt it was an acceptable surrogate for estimating drug target expression in various tumor models. Ref 26.

"No associations were found between EFS and PARP1 mRNA expression (**Figure 2a**) or between genetic sub-groups (**Figure 2b**)."

211At-MM4 was previously shown to be effective in PARP-expressing cancer models. What is added value here? Only xenograft -> PDX .

[211At]PTT efficacy has been reported for 1 established human xenograft model and results showed impressive response. By testing 11 additional PDX models that were complete with genetic characterization we identified that some PDX models are quite recalcitrant to therapy (like the cancers from which they derived) and showing efficacy is very optimistic for human translation.

Is an MTD relevant for TAT therapy? It seems the lower dose regimen may have a better therapeutic index.

While a cytotoxic treatment with efficacy below MTD would be a major goal of cancer treatment, in relapsed/refractory disease we would still escalate to MTD as a first step with de-escalation only if we achieve high rates of durable remission.

What was the uptake of 211At-MM4 in NB tumors?

Uptake was not assessed in models due to the high number of PDX models evaluated.

PARP expression?

Referenced here.

“Patient demographics and genetic characteristics are listed in (SI Tables 1 and 2) (9).”

“No associations were found between EFS and PARP1 mRNA expression (Figure 2a) or between genetic sub-groups (Figure 2b).”

“Although PARP1 protein expression was not directly evaluated, PARP1 mRNA has been shown to be highly correlated to PARP1 protein expression justifying our approach (26).”

Dosimetry?

Tumor dosimetry was not performed.

Given the late-stage weight loss in some animals, was the cause investigated?

No, the cause was not investigated.

Were there late toxicities?

Late toxicities are not required for clinical translation.

“A major limitation of the current pre-clinical trial design is that late toxicities of [211At]PTT were not directly assessed in a healthy mouse model, however this does not limit clinical translation to First-In-Human phase 1 and phase 2 clinical trials under FDA guidelines that recommend single species long term toxicity studies at the time of marketing “Oncology Therapeutic Radiopharmaceuticals: Nonclinical Studies and Labeling Recommendations Guidance for Industry”. ”

Was histology or other damage in normal tissue evaluated?

See above

Fig 1: Are any control animals included, to probe tumour growth in case of no treatment (fig 1c)? It is impossible to gauge how large tumors are at the start of therapy? EFS for control animals seems very short?

Control animals are not listed in Figure 1 c. Control tumors progressed from 0.2 cm³ to >2cm³ within approximately 20 days for all models. This means all control animals would have a 1000% increase in

tumor growth. We felt this would be un-helpful to show, despite how impressive it makes the agent under evaluation appear.

SI Fig 2: were only 2 animals per group treated with 211At per PDX?

Yes, only 2 animals per group were treated with [211At]PTT for each respective dose level.

Although therapeutic affect is observed across the PDXs, does this allow statistical analysis?

Yes, statistical tests were performed under conditions that tested for differences in therapeutic effect of [211At]PTT vs. saline control in tumors. No conclusions were made regarding individual models, although several models were treated at multiple dose levels and significant differences were observed for treated vs. untreated groups within models with ≥ 2 dose levels evaluated.

This has been added as SI Figure 3.

“Grouped analysis for models treated at all dose levels (COG-n-415x, COG-n-440x, and COG-n-519x) showed significantly longer EFS compared to controls (T-test, p-value < 0.05) (SI figure 3).”

What is the relevance of the different PDX models, if no conclusion can be drawn regarding the individual PDX models?

Testing multiple models is more relevant to understanding if a drug would be broadly effective in a specific cancer type. A single model only represents a single patient at best, and more realistically a single tumor cell lineage. Testing 11 models increases the significance of the work by establishing a generalizable outcome for a given tumor type such as Neuroblastoma evaluated here.

The publication that is referenced to justify n=2, regards drug screening, not multiple model screening. The relevance of this reference, and the choice of n=2 requires additional justification.

The paper describes a study design to enable rapid screening of drugs across multiple PDX cancer models and is intended to rapidly evaluate a single drug across multiple models.

Tumor rechallenge in only 2 animals is anecdotal at best. How much conclusion can be drawn from 2 animals?

Regarding the sample size, we do not make any claims about the statistical significance of tumor re-challenge, however observations are objectively reported. For an individual animal with re-challenge we calculate event free survival by method of area over the curve. The conclusion we made is that the dose response in this single animal was nearly identical upon re-treatment and occurred in two unique subjects from different models.

It seems mouse weight was considerably affected by 211At-PTT (SI Fig 2C). Was mouse weight loss

dependent on administered dose? Some animals seem to lose >20% of their body weight? Was this corrected for tumor size?

Mouse weight loss was dose dependent and mice that dropped below 20% are discussed here. This was corrected for tumor size.

Stated here

“Two out of 22 (9%) of mice treated at the MTD were censored from EFS analysis due to weight loss >20%, albeit >50 days from treatment and both subjects had no measurable disease at time of removal (NB-1643x at 76 days, and COG-N-471x at 65 days). One dose limiting toxicity was observed in an NE-EBC1x tumor bearing mouse re-challenged with 24 MBq/kg administered as a fractionated regimen (**Figure 1f**), and overall 43 of 46 treated mice for all dose levels maintained a healthy weight throughout the study ($\geq 100 \pm 20\%$ body weight) (mouse weights shown in **SI Figure 2c**).”

And here

“The MTD in CB57/BL6 mice was determined to be 48 MBq/kg/fraction administered twice weekly for a total of 4 dose fractions, which is higher than the MTD for CB57 SCID mice (**SI Figure 3a**).

This statement was added for clarification

“Weight loss toxicity was dose dependent in both mouse models.”

How is thyroid uptake explained? Is this an issue (given the radiation dose).

Thyroid uptake could be due to PARP1 expression or deastatination. PARP1 is highly expressed in sub-populations of thyroid cells. Ref 28.

What is the radiation dose to the tumor?

Radiation dose to tumor was not calculated in this study and is expected to be variable for each tumor model based on the variable PARP1 mRNA expression.

Is there de-astatination? Is tumor uptake specific (blockable)?

Deastatination is possible and tumor specific uptake was not assessed. However, previously we showed free astatine is <100 times as potent as [211At]PTT in vitro therefore we would expect reduced efficacy in vivo. Studies are ongoing to determine in vivo kinetics and deastatination.

Reviewers' comments:

Reviewer #1 (Remarks to the Author):

Following revision, this manuscript has been considerably improved and the most of the questions raised have been adequately answered.

I have one remaining comment: In my previous review I asked for a reference to support the statement that nuclear delivery is superior to targeted extracellular delivery of alpha-emitting radionuclides. While the theoretical concept is compelling, I was specifically interested in any evidence of an actual clinical benefit of this approach. I acknowledge that such evidence might not currently exist and that a requirement for such evidence might fall outside of the scope of this paper. However, the reference made to Zhu et al 2017 seems slightly misleading. The statement that only 0.9% of ²²⁵Ac-DOTA was associated with cells, is taken out of its original context: DOTA is only a chelator, not a targeting molecule and ²²⁵Ac-DOTA, correctly referred to as "non-targeting" by Zhu et al., was not expected to associate with the cells. The second part of the sentence "...and did not appear to penetrate cells at all." is a more relevant, albeit not particularly strong support of the chosen approach.

Reviewer #2 (Remarks to the Author):

The revisions to the previous manuscript have significantly improved this original and important paper.

Mark Gaze

Reviewer #3 (Remarks to the Author):

Off-target toxicities and de-astatination should be addressed.

Justification for 2 animals per model should be fully justified.

Dosimetry should be added.

Response to Reviewers' comments:

Reviewer #1 (Remarks to the Author):

Following revision, this manuscript has been considerably improved and the most of the questions raised have been adequately answered.

I have one remaining comment: In my previous review I asked for a reference to support the statement that nuclear delivery is superior to targeted extracellular delivery of alpha-emitting radionuclides. While the theoretical concept is compelling, I was specifically interested in any evidence of an actual clinical benefit of this approach. I acknowledge that such evidence might not currently exist and that a requirement for such evidence might fall outside of the scope of this paper. However, the reference made to Zhu et al 2017 seems slightly misleading. The statement that only 0.9% of ²²⁵Ac-DOTA was associated with cells, is taken out of its original context: DOTA is only a chelator, not a targeting molecule and ²²⁵Ac-DOTA, correctly referred to as "non-targeting" by Zhu et al., was not expected to associate with the cells. The second part of the sentence "...and did not appear to penetrate cells at all." is a more relevant, albeit not particularly strong support of the chosen approach.

Response: We have removed the statement "In fact, only 0.9% of ²²⁵Ac chelated with dodecane tatraacetic acid (²²⁵Ac-DOTA) was shown to associate with cells and did not appear to penetrate cells (16)." and included this statement for clarity "Although the question of clinical significance of targeting alpha-emitters to the nucleus remains to be unanswered, in theory it should have beneficial effects especially for cancer specific targets."

Reviewer #2 (Remarks to the Author):

The revisions to the previous manuscript have significantly improved this original and important paper.

Mark Gaze

Reviewer #3 (Remarks to the Author):

Off-target toxicities and de-astatination should be addressed.

Response: Biodistribution of [²¹¹At]PTT in an NBEBEC1 tumor bearing model was added. De-astatination was shown to be insignificant for tumor uptake at 1 hour. Furthermore we conclude that deastatination is not a concern for off-target toxicity, since the MTD of free-astatine-211 is >50 MBq/kg in mice.

Justification for 2 animals per model should be fully justified.

Response: For the comment about the N=2 design: The Maris lab has performed preclinical trials of hundreds of novel therapeutics over the last two decades through the NCI funded Pediatric Preclinical In Vivo Testing Program (PIVOT: <https://preclinicalpivot.org/>). Through these efforts we have adopted "single mouse testing" to more closely recapitulate a clinical trial (ref: 10.1158/0008-5472.CAN-16-0122). We use this design when we expect a large effect size and have sufficient diversity in xenograft models to explore hypotheses such as biomarkers of response or resistance. We often modify to a N=2

design if the mechanisms of synthesis or delivery are complicated (ref: 10.1158/1078-0432.CCR-20-4221). Single mouse or N=2 are now commonly deployed through the PIVOT and Maris lab preclinical testing programs.

Dosimetry should be added.

Response: Tumor dosimetry was added for an NB-ECB1x tumor model. See SI Figure 5 and SI Table 5.

REVIEWERS' COMMENTS:

Reviewer #3 (Remarks to the Author):

No further comments